# A Novel *Lactobacillus acidophilus* Strain Isolated from a 2-Month-Old Shiba Inu: In Vitro Probiotic Evaluation Safety Assessment in Mice and Whole-Genome Sequencing Analysis

**DOI:** 10.3390/microorganisms13092095

**Published:** 2025-09-08

**Authors:** Huiming Huang, Xiaoling Tang, Yichuan Zhang, Mengyao Chen, Min Wen

**Affiliations:** 1Shandong Key Laboratory of Applied Technology for Protein and Peptide Drugs, School of Pharmaceutica Sciences and Food Engineering, Liaocheng University, Liaocheng 252059, China; hmhuang1988@163.com (H.H.); 19563562338@163.com (X.T.); 2College of Agriculture and Biology, Liaocheng University, Liaocheng 252059, China; z13563349103@126.com (Y.Z.); 15964650175@163.com (M.C.); 3Shandong Key Laboratory of Applied Technology for Protein and Peptide Drugs, Institute of Biopharmaceutical Research, Liaocheng University, Liaocheng 252059, China; 4Pet Nutrition Research and Development Center, Gambol Pet Group Co., Ltd., Liaocheng 252000, China

**Keywords:** *Lactobacillus acidophilus*, probiotic properties, feces, 2-month-old Shiba Inu, safety evaluation, whole genome sequencing

## Abstract

Owing to their remarkable biological activities and health benefits, probiotics have gained widespread application in enhancing pet health and welfare. Host-derived probiotics are considered optimal due to their unique digestive tract environments. This study isolated *Lactobacillus acidophilus* L1 from the feces of a 2-month-old Shiba Inu puppy and conducted a comprehensive evaluation of its potential as a probiotic candidate for pet health. Strain L1 demonstrated high tolerance to acidic conditions (survival rates of 90.41%, 92.90% and 98.81% at pH 2, 2.5, and 3.0, respectively) and bile salts (survival rates of 98.05%, 95.68%, and 82.21% at 0.1%, 0.2%, and 0.3% concentrations, respectively). Adhesion to Caco-2 intestinal epithelial cells reached 38.33%, with hydrophobicity of 97.81% and auto-aggregation of 32.28%. L1 also displayed pronounced antioxidant activity, with DPPH and ABTS radical scavenging rates of 71.15% and 83.20%. Both the bacterial suspension and the cell-free supernatant had potent inhibition of pathogenic bacteria, while the strain showed a non-hemolytic phenotype and remained sensitive to clinically relevant antibiotics (e.g., penicillin). On the other hand, animal experiments conducted in ICR mice (randomly divided into four groups) demonstrated that oral administration of L1 had no toxic effects on the mice and increased serum SOD and CAT levels, while reducing MDA levels. Furthermore, whole-genome sequencing revealed that L1 is 2,106,895 bp in size and contains 2098 coding sequences, two CRISPR arrays, ten genomic islands, and two prophage regions. Collectively, the in vitro and in vivo data presented here indicate that *L. acidophilus* L1, originally isolated from canine feces, supports further evaluation as a candidate strain for incorporation into functional pet foods.

## 1. Introduction

Dogs have coexisted with humans for thousands of years. As pet owners increasingly regard their pets as integral family members, pet health has become a paramount concern [1]. The canine gastrointestinal (GI) microbiota plays a crucial role in maintaining and improving canine health by influencing digestion, producing key metabolites, and modulating the immune system [2]. The composition of the gut microbiota is influenced by numerous factors, including age, nutrition, and environment [3,4]. Among these, age exerts one of the most significant impacts on microbial composition [5]. During early development, the gut microbiota undergoes rapid succession, as key anaerobic bacteria progressively colonize the intestine prior to adulthood [6,7]. These bacterial communities are of vital importance for the host health, as disturbances in their composition can lead to altered metabolic states, such as acute diarrhea or inflammatory bowel disease [8,9]. During early development, the gut microbiota undergoes rapid succession, as key anaerobic bacteria progressively colonize the intestine prior to adulthood.

Probiotics offer an effective means of regulating intestinal microbiota and promoting overall health [10,11]. *Lactobacillus acidophilus*, a homofermentative, slightly aerobic, Gram-positive bacterium within the genus *Lactobacillus* [12], is widely recognized as a safe and effective probiotic. Studies have demonstrated that *Lactobacillus acidophilus* can inhibit pathogenic microorganisms, enhance the intestinal epithelial barrier, reduce pro-inflammatory factors, and support other immune functions [13,14,15,16]. Host-derived probiotics are considered optimal because of the unique digestive tract environments and host species specificity of probiotics across different animals [17,18]. Furthermore, research has demonstrated that the fecal microbiota of puppies at 8 weeks of age remains significantly different from that of their dams [19]. However, the majority of probiotics currently available in the pet market are non-canine strains [20], and research on probiotics for puppies is even more limited. Moreover, most of these strains have not been adequately evaluated for their probiotic properties, resulting in inconsistent outcomes. These limitations prevent them from meeting the growing demands of the rapidly developing pet market [21]. Therefore, there is an urgent need to explore probiotic resources of lactobacilli from puppy sources.

In this study, a strain of *Lactobacillus acidophilus* (designated L1) was isolated from the gastrointestinal tract of a 2-Month-Old Shiba Inu Puppy. Its probiotic potential and safety were systematically evaluated through in vivo and in vitro assays, complemented by whole-genome sequencing. The findings expand the microbial resource pool of puppy-derived lactic acid bacteria and provide theoretical support for the development and functional application of probiotics in companion animals.

## 2. Materials and Methods

### 2.1. Sample Collection

A fecal sample was obtained from a healthy 2-month-old Shiba Inu from Pet Nutrition R&D Center, Gambol Pet Group Co., Ltd. in Liaocheng, China. The test animal had not been administered antibiotics or probiotics prior to sample collection. Samples were collected from the rectum using sterile swabs and were promptly placed in sterile test tubes, and transported under refrigeration to the laboratory.

### 2.2. Strain Isolation and Identification

Take about 1 g of fresh feces, dilute it with sterile PBS buffer at a ratio of 1:9, and remove the impurities in the feces by filtration using sterile gauze. The supernatant was diluted according to the ratio of 1:1000, and 100 µL of the supernatant was evenly spread on MRS solid medium and incubated anaerobically at 37 °C for 24~48 h. Individual colonies were selected for passaging on MRS agar and then incubated for 48 h at 37 °C. After three passages, the isolates were stored in 50% (*w*/*v*) sterile glycerol at −80 °C. The morphology and color of bacteria were observed by Gram staining.

To identify the isolate, genomic DNA was extracted using a DNA extraction kit (Solabio, Beijing, China). Polymerase chain reaction (PCR) amplification was carried out using the primers 16S-27F (5′-AGAGTTTGATCCTGGCTCAG-3′) and 16S-1492R (5′-TACGGCTACCTTGTTACGACTT-3′) [22]. The PCR products were identified by 1.0% agarose gel electrophoresis, and the target bands were extracted using a gel extraction kit (Omega Bio-Tek, Norcross, GA, USA) and sent to Qingdao Weilai Biotechnology Co. (Qingdao, China) for sequencing. A phylogenetic tree was generated using Mega 7.0 software to determine the evolutionary relationships of the isolate [23].

The present study reports the isolation and identification of *Lactobacillus acidophilus* L1. The strain has been preserved in the China Center for Type Culture Collection, under the designation CCTCC NO. 31008.

### 2.3. Evaluation of Probiotic Properties

#### 2.3.1. Growth and Acid-Producing Ability

An inoculum of 2.0% (*v*/*v*) of the strain was used to inoculate MRS broth, which was then incubated at 37 °C for 48 h. During anaerobic incubation at 37 °C, samples were collected every 2 h to measure the OD 600 nm and pH.

#### 2.3.2. Acid and Bile Salt Resistance

The method described by Jomehzadeh et al. was followed with minor modifications [24]. Acid resistance test: The pH of the MRS liquid medium was adjusted to 2.0, 2.5, and 3.0 with 0.5 mol/L HCl solution and was checked again after sterilization. A total of 200 µL of fermentation broth was added to 4.8 mL of MRS liquid medium of different pH values and incubated anaerobically at 37 °C. Samples were taken at 0 and 3 h, diluted to the appropriate concentration and spread evenly on solid MRS plates. After 24 h of incubation, the number of colonies on the plates was recorded and the survival rate was calculated.Survival rate (%) = N3/N0 × 100%(1)

N3 is the number of surviving bacteria after 3 h; N0 is the number of surviving bacteria after 0 h; the number of surviving bacteria is expressed as lgCFU/mL.

Bile salt tolerance test: 200 µL of the fermentation broth was added to 4.8 mL of MRS substrate containing 0.1%, 0.2% and 0.3% (*w*/*v*) concentrations of bile salts and incubated anaerobically at 37 °C. Samples were taken at 0 and 3 h, respectively. The bile salt tolerance of the strains was calculated using the same plate dilution method as in the acid tolerance test.

#### 2.3.3. Antimicrobial Ability

The antimicrobial ability of the strain was assessed utilizing the Oxford cup double-layer agar diffusion technique [25]. The strain L1 was inoculated into MRS medium and incubated anaerobically at 37 °C for 18 h. The bacterial suspension (BS) concentration was adjusted to approximately 1 × 10^9^ CFU/mL using MRS medium. The bacterial suspension (BS) was followed by centrifugation at 10,000 rpm for 10 min at 4 °C to collect the cell-free supernatant (CFS), and the remaining bacterial pellet (BP) was washed three times with PBS (pH 7.0) and resuspended in the same volume [26]. In addition, indicator bacteria (Methicillin-resistant *Staphylococcus aureus* CCARM 3090; *Escherichia coli* CCARM 1009; *Salmonella typhimurium* CCARM8250; *Staphylococcus aureus* CMCC26003; *Klebsiella pneumoniae* ATCC10031) were inoculated into Lysogeny broth and incubated at 37 °C for 18 h, after which the cell density was adjusted to approximately 1 × 10^8^ CFU/mL. Indicator bacteria were spread on LB agar plates, which had 6 mm wells punched using Oxford cups. Each well was filled with 100 μL of BS, CFS, BP and MRS broth. The samples were then incubated at 37 °C for 24 h, after which the plates were observed to measure the diameter of the inhibition zones (IZD). Each test was repeated three times.

### 2.4. Safety Assessment of the L1 Strains

#### 2.4.1. Hemolytic Activity

After overnight incubation (18 h), cultures were incubated on Columbia CNA Blood Agar medium for 24 h at 37 °C. *Staphylococcus aureus* CMCC26003 was used as a positive control. Hemolysis was assessed based on the appearance of hemolytic zones: α-hemolysis (greenish discoloration), β-hemolysis (clear, complete lysis), and γ-hemolysis (no hemolysis) [27].

#### 2.4.2. Antibiotic Susceptibility

Strain susceptibility was determined by the paper agar diffusion method. The concentration of fresh culture was adjusted to 2 × 10^6^ CFU/mL, 100 µL of bacterial solution was deposited uniformly on the surface of MRS solid medium and left at room temperature for approximately 10 min, antibiotic disks were placed on the plate surface and incubated anaerobically at 37 °C for 24 h, then the diameter of the transparent circle on the susceptible pellets was measured and recorded using a caliper. Inhibition zone diameters (IZD), inclusive of the disk diameter, were measured. Based on IZD values, isolates were classified as sensitive (≥21 mm), intermediate (16–20 mm), or resistant (≤15 mm) [28].

#### 2.4.3. Autoaggregation Activity and Cell Surface Hydrophobicity

Referring to the test method of Zou et al. [29], the fermentation broth of the overnight cultured strain was centrifuged (6000 rpm, 10 min) to obtain the cell, which was washed twice with PBS and the concentration of the bacterial solution was adjusted with PBS to the OD600 nm of approximately 0.6 ± 0.05 (A_0_) for standardization. The bacterial suspension with adjusted concentration was vortexed for 10 s. The supernatant absorbance at 600 nm was measured (A_t_) after 1, 3 and 5 h at 37 °C. Subsequently, an equal volume of xylene, ethyl acetate, chloroform and the bacterial suspension were vortexed to mix well, placed at room temperature for 30 min, and then the aqueous phase was carefully aspirated to measure the OD600 nm and recorded as A_1_.Auto aggregation activity (%) = (A_0_ − A_t_)/A_0_ × 100%(2)Cell Surface Hydrophobicity (%) = (A_0_ − A_1_)/A_0_ × 100%(3)

### 2.5. Adhesion to Human Colon Carcinoma (Caco-2) Cells

Bacterial adhesion was evaluated with Caco-2 cell monolayers cultured in high-glucose DMEM (Dulbecco’s Modified Eagle Medium) containing 20% (*v*/*v*) FBS (fetal bovine serum), 100 μg/mL penicillin, and 100 μg/mL streptomycin. Caco-2 cells were cultured at 37 °C in a thermostat incubator containing 95% air and 5% carbon dioxide.

#### 2.5.1. FITC Labeling of Test Strains

FITC (fluorescein isothiocyanate) solution was prepared at a concentration of 500 μg/mL by dissolving 0.5 g FITC in 1 mL DMSO, followed by a 1000-fold dilution. Bacterial cells were washed three times with PBS (pH 7.0) and resuspended in the FITC solution at 1 × 10^7^ CFU/mL. The suspension was incubated at 37 °C with shaking for 2 h in the dark. After incubation, the organisms were washed three times with PBS to remove unbound FITC solution, resuspended in 20% DMEM cell culture medium without antibody and adjusted to a final concentration of 2 × 10^8^ CFU/mL, and the relative fluorescence intensity values (RFU) of the strains were determined at a wavelength of 485 nm (absorption wavelength) and a wavelength of 530 nm (emission wavelength), which were recorded R_0_ as the relative fluorescence intensity value of the strains prior to the adherence of the strain.

#### 2.5.2. Cell Adhesion Assay

Caco-2 cells were cultured in 24-well plates at 37 °C under 5% CO_2_ until reaching confluence. (The Caco-2 cell line used in this study was kindly provided by Professor Hongzhuan Xuan’s research team (Liaocheng University, Shandong, China). This cell line traces back to the original ATCC repository with the accession number HTB-37.) The monolayers were then washed three times with phosphate-buffered saline (PBS, pH 7.0), and 600 µL of labeled strain solution added. The cells were incubated from light for 2 h. The culture solution was discarded and washed with PBS three times to remove the strains that had not adhered to the cells. Each well was digested by adding 300 µL 0.25% trypsin for 5 min, and the digestion was terminated with 300 µL DMEM cell culture medium. The relative fluorescence intensity value (RFU) cell suspension was measured under the same wavelength condition and recorded as R.Cell Adhesion rate % = R/R_0_ × 100%(4)

#### 2.5.3. Confocal Microscopy Observation

The following experiments were performed under light-protected conditions. Caco-2 cells were added to glass bottom cell culture dishes at 5 × 10^5^cell/mL and incubated for 24 h, the cell supernatant was aspirated and washed three times with PBS; FITC-labeled strains were added to the dishes and incubated for 1 h, and then washed three times with PBS to remove the unadhered strains; 1 mL of 4% paraformaldehyde was added and left at room temperature for 30 min; the fixative was aspirated and washed three times with PBS; the Actin-Tracker Red-594 was diluted with PBS containing 5% BSA at a ratio of 1:100; subsequently, 1 mL of the diluted solution was added to each dish. The samples were then incubated at room temperature in the dark for 10 min to stain the cytoskeleton after the cells were washed three times with PBS, and excess liquid was carefully removed. Finally, 20 µL of anti-fluorescence quencher was added, and the samples were observed under a laser confocal microscope, with multiple fields randomly selected for analysis.

### 2.6. The Ability to Scavenge 2,2-Diphenyl-1-Picrylhydrazyl (DPPH) and 3-Ethylbenzthiazoline-6-Sulfonic Acid (ABTS) Radicals

Radical scavenging capacities against DPPH and ABTS were assessed using specific reagent kits (Solarbio, China) for each assay [30]. The absorbance of the resulting solution was measured at OD515 nm or OD405 nm.DPPH/ABTS radical scavenging rate (%) = [[Ablank − (Aassa − Acontrol)]/Ablank] × 100%(5)

### 2.7. Safety Evaluation of the Strain L1 in Mice

Forty SPF-grade ICR mice (4 weeks old, 20–23 g, equal numbers of males and females) were obtained from Pengyue Laboratory Animal Breeding Co., Ltd. (Jinan, China) and randomly assigned to four groups of ten animals each. Following a 7-day acclimatization period under controlled conditions (25 ± 1 °C; ad libitum access to food and water), the oral gavage regimen was initiated. The control group (C) received daily oral gavage of 0.2 mL sterile saline for 28 days, while low- (L), medium- (M), and high-dose (H) groups were administered 0.2 mL of Strain L1 suspensions at 1 × 10^7^, 1 × 10^8^, and 1 × 10^9^ CFU/mL, respectively [31]. Throughout the study, mice were monitored daily for clinical signs (including fecal consistency) and weighed weekly. Prior to terminal sampling, animals were fasted for 12 h, anesthetized via intraperitoneal injection of 1.0% (*w*/*v*) sodium pentobarbital (50 mg/kg), and exsanguinated via abdominal aorta puncture. Blood samples were centrifuged at 4000× *g* (4 °C, 15 min) to obtain serum aliquots. Subsequently, the collected serum was analyzed for various biochemical markers utilizing commercial assay kits, which were procured from the Jiancheng Bioengineering Institute in Nanjing, China. Following euthanasia, mice were subjected to necropsy with gross examination of visceral organs. The liver, spleen, kidneys, and heart were excised, weighed, and the relative organ weight (organ-to-body weight ratio) was calculated as: (organ weight/final body weight) × 100% [32]. Additionally, histopathological analyses were conducted on liver, spleen and kidney from mice in the high-dose group.

### 2.8. Whole-Genome Sequencing

The genome was sequenced using the PacBio and Illumina platforms at Majorbio Bio-Pharm Technology Co., Ltd. (Shanghai, China). De novo assembly was performed with Unicycler (v0.4.8) to generate a high-quality genome [33]. The genes were analyzed with Clusters of Orthologous Groups (COG), Kyoto Encyclopedia of Genes and Genomes (KEGG), and Gene Ontology (GO). Virulence factors were annotated against the VFDB database, applying a coverage threshold of >60%. Clustered regularly interspersed short palindromic repeats (CRISPR) were identified by Minced (Version: 0.4.2) [34]. The final assembled genome was submitted to the NCBI database (accession number PRJNA1262664).

### 2.9. Statistical Analysis

Data are presented as mean ± standard deviation (SD). Statistical analyses were performed using GraphPad Prism 10.3.1, employing Student’s *t*-test, one-way ANOVA, and two-way ANOVA where appropriate. Inter-group differences were considered statistically significant at *p* < 0.05.

## 3. Results

### 3.1. Isolation and Morphological Observation of Strains

The strains were isolated from MRS Solid medium and named L1. As can be seen from Figure 1a, the colonies on MRS solid medium were all white round single colonies with smooth surfaces. The strain was identified as Gram-positive by Gram staining and showed purple rod shape (Figure 1b).

### 3.2. Molecular Biological Identification of the Strain

The electrophoretic detection of PCR amplification products is shown in Figure 1c, and the size of the band was around 1500 bp. The phylogenetic tree was constructed using BLAST+2.16.0 homology composite score, and Figure 1d shows that the sequence homology of the strains with the reported *Lactobacillus acidophilus* was greater than 99.8%. Therefore, it was determined that the strain L1 belonged to *Lactobacillus acidophilus*.

### 3.3. Antimicrobial Ability

The inhibitory ability of L1 strain against five pathogenic indicator bacteria: The most significant inhibitory effect of CFS was *S. aureus*, followed by *S. typhimurium*, MRSA, *K.pneumoniae* and *E. coli*. However, neither BS nor CFS exhibited significantly different inhibitory effects against the same pathogenic bacteria. Furthermore, the BP of L1 had no inhibitory effect on pathogenic bacteria (Figure 2).

### 3.4. Functional Characteristics of L1

#### 3.4.1. Growth and Acid-Producing Ability Curves

It can be seen from Figure 1e that the strain was in a lag period in the first 2 h, and the pH of the medium changed little; the growth of the strains accelerated at 2~10 h, entering the logarithmic growth period, and the pH decreased from 6.9 to about 5.23; the strain concentration did not change significantly after 10 h, but still had an increasing trend, and reached the maximum growth at about 18 h; its pH continued to decrease after 10 h, and the pH was basically stable about 4.0 after 18 h.

#### 3.4.2. Tolerance to Different Conditions

The results display the survival rates for acid and bile salt resistance (Figure 3a,b), L1 survival remained over 90% at pH 2.0 and was as high as 98% at pH 3.0. The data demonstrate that 0.1% bile salt exerted minimal impact on L1 survival, whereas exposure to 0.3% bile salt resulted in a significant reduction in viability to 82% relative to control. The study demonstrated the good viability of L1 under both gastric and intestinal conditions.

#### 3.4.3. Cell Surface Hydrophobicity and Autoaggregation Activity

The strain L1 has different degrees of hydrophobicity towards three organic solvents: xylene, trichloromethane and ethyl acetate, among which the hydrophobicity towards the trichloromethane is the largest at 97.82%, xylene is 78.44%, ethyl acetate is the lowest at 17.09% (Figure 3c). The autoaggregation capacity of L1 increased with time, up to 32.28% at 5 h (Figure 3d).

#### 3.4.4. In Vitro Analysis of Antioxidant Activity

Figure 3e shows the antioxidant activity of the L1 strains. *L. acidophilus* L1 demonstrated potent antioxidant capacity, albeit with significantly lower DPPH radical scavenging activity (71.15%) compared to ABTS radical scavenging (83.20%).

#### 3.4.5. Adhesion to Caco-2 Cells

Caco-2 cells were used to determine the adhesion of probiotic strains, and the adhesion of L1 was 38.33% (Figure 3f); the cytoskeleton was stained with Actin-Tracker Red-594 to observe probiotic adhesion under a laser confocal microscope, with a scale bar equal to 80 μm. The green fluorescence is the FITC-labeled probiotic strain, the red is the labeled cytoskeleton, and the probiotic adheres to the surface of the Caco-2 cells, which can be clearly seen (Figure 3i).

### 3.5. Safety Assessment of L1

#### 3.5.1. Hemolytic Activity

*L. acidophilus* L1 exhibited a γ-hemolytic (non-hemolytic) phenotype, indicating its safety potential. Representative results are presented in Figure 3g,h.

#### 3.5.2. Antimicrobial Susceptibility

To assess biosafety, the antibiotic susceptibility profile of *L. acidophilus* L1 was evaluated against ten antimicrobial agents. As demonstrated in Table 1, L1 exhibited susceptibility to penicillin, ampicillin, chloramphenicol, cefazolin, and erythromycin, while demonstrating resistance to the remaining five antibiotics.

### 3.6. Safety Evaluation In Vivo

Safety experiments were conducted for oral administration of L1 in mice. During the experiment, all mice remained active, with no cases of diarrhea, death, or other signs of illness observed. Autopsy results revealed no significant pathological changes in various organs. The body weights of mice treated with L1 across all three groups exhibited dose-dependent reductions compared to the control group, with greater weight loss observed at higher doses (Figure 4a). However, these differences did not reach statistical significance among the treatment groups (*p* > 0.05). Additionally, measurements of heart, liver, spleen, and kidney coefficients in the L1-administered mice, showed no significant differences compared to the control group (Figure 4b–f). Liver and kidney function biomarkers were assessed in all mouse cohorts. No significant differences (*p* > 0.05) were observed in serum levels of blood urea nitrogen (BUN), creatinine (CRE), aspartate aminotransferase (AST), alanine aminotransferase (ALT), and triglycerides (TG) between the *L. acidophilus* L1-treated group and the control (Figure 4g–k). However, the medium- and high-dose L1 supplementation groups showed significantly elevated activities of superoxide dismutase (SOD; *p* < 0.05) and catalase (CAT; *p* < 0.05) compared to the control group (Figure 4l,n). In contrast, glutathione peroxidase (GSH-Px) activity remained unchanged (Figure 4m). Furthermore, malondialdehyde (MDA) levels were markedly reduced (*p* < 0.001) in high-dose L1-administered mice relative to the control group (Figure 4o). Following the administration of a high dose of L1 orally, histopathological examination showed no abnormalities in the liver, spleen, or kidneys (Figure 4p).

### 3.7. Whole-Genome Sequencing and Bioinformatics Processing

Whole-genome sequencing was conducted to thoroughly characterize the probiotic strain and to further explore the capabilities of L1. The complete circular genome map of L1 is shown in Figure 5. The complete genome of L1 comprises one 2.107 Mbp circular chromosome and one circular plasmid, with guanine and cytosine (G + C) contents of 34.97 and 36.10%, respectively. Table 2 shows the genomic information of L1, the genome contains a total of 2098 genes with an average length of 879.93 bp, and the total length of the gene sequences is 1,846,086 bp, accounting for 87.62% of the total genome length. Additionally, the genome contained 260 pseudogenes, 2 CRISPR arrays, 10 gene islands, and 2 prophages.

A total of 90 carbohydrate-active enzymes (CAZymes) were predicted, including 44 glycoside hydrolases (GHs) and 22 glycosyltransferases (GTs) (Table 3). Figure 6a illustrates the genomic map of annotated CDSs in L1 based on general databases. A total of 1677 genes were successfully annotated to the COG database, representing 79.93% of the total gene count. These genes were classified into 23 functional categories. The category with the highest number of genes was Translation, ribosomal structure and biogenesis (J), with 190 genes (Figure 6b). The genome contains 1389 genes that can be annotated to the GO database, representing 66.21% of the total number of genes. As depicted in Figure 6c, GO annotation revealed that: Biological Process was predominantly associated with translation; Cellular Component was primarily linked to integral membrane components; Molecular Function was chiefly related to ATP binding. In the whole genome, a total of 1528 genes were successfully annotated to the KEGG database, representing 72.83% of the total gene count. These genes were classified into 40 distinct KEGG pathways, the most gene-rich pathways identified were global and overview maps, followed by carbohydrate metabolism, membrane transport and amino acid metabolism (Figure 6d). A total of 120 antibiotic resistance genes were identified in the CARD database (Figure 7a). Based on selection criteria of >80% sequence identity and >80% coverage, two genes—*lnuC* (gene 1087) and *tetW* (gene 0057)—were chosen from the CARD and ResFinder databases (Appendix A). Annotation of the VFDB database identified 181 virulence factor-related genes, with seven genes meeting thresholds of >60% sequence identity and >80% coverage (Figure 7b, Appendix A). Annotation revealed 454 pathogen-host interactions (Figure 7c) and 390 membrane transporter proteins, with primary active transporters predominating (Figure 7d). Additionally, the L1 genome encodes genes involved in bile-salt hydrolase, bacteriocin production, and adhesion (Table 4).

## 4. Discussion

Probiotics are defined “living microorganisms that, when consumed in sufficient quantities, have beneficial effects on the health of the host” [35]. To qualify as potential probiotics, candidate strains must possess the ability to withstand acidic conditions and bile salts [36]. It has been demonstrated that different strains exhibit varying degrees of tolerance to the harsh conditions of the animal gastrointestinal tract [37]. Therefore, the environmental tolerance of strains is a pivotal factor in their evaluation. Zhao et al. [30] isolated *Pediococcus acidilactici* GLP02 and GLP06 from adult beagle feces, and after treatment with culture solution of pH 2.5 for 3 h, the survival rates of strains GLP02 and GLP06 were 63.97% and 72.17%, respectively. Additionally, their survival rates in the presence of 0.30% bile salts were 98.84% and 95.70%, respectively. By comparison, *L. acidophilus* L1, isolated in this study, exhibited a survival rate of 92.90% after 3 h at pH 2.5 and 82.81% after 3 h in 0.3% bile salts. These findings indicate that strain L1 possesses markedly superior acid tolerance, although its bile salt tolerance is slightly lower than that of GLP02 and GLP06, likely reflecting differences in strain origin.

For probiotics to exert sustained effects, they must not only survive passage through the gastrointestinal tract but also colonize and multiply within it. Auto-aggregation activity and Cell surface hydrophobicity are key characteristics used to assess a strain’s capacity for gastrointestinal colonization [38]. Xing et al. [15] demonstrated that *L. acidophilus* AD125 exhibited an auto-aggregation rate of 26.51% and a hydrophobicity of 93.45% with xylene, indicating strong adhesion capacity. In comparison, *L. acidophilus* L1 showed a higher auto-aggregation rate (32.28%) and hydrophobicity (97.81% with chloroform), suggesting superior colonization potential. Beyond colonization, functional activities such as antioxidant and antimicrobial properties further enhance the probiotic value of a strain. These traits may contribute to the prevention of oxidative stress–related diseases and reduce the risk of inflammation and carcinogenesis [39,40]. Notably, *L. acidophilus* L1 demonstrated strong free radical–scavenging activity against DPPH and ABTS, underscoring its considerable antioxidant capacity.

Moreover, metabolites produced by *Lactobacillus*, including organic acids and bacteriocins, are known to inhibit pathogenic bacteria [41]. In the strain inhibitory activity test, the bacterial suspension (BS) and cell-free supernatant (CFS) of L1 effectively suppressed the growth of pathogenic bacteria, whereas the bacterial pellet (BP) showed no effect, indicating that antimicrobial activity is mainly mediated by strain-derived metabolites. Because safety is strain-specific, each candidate probiotic must undergo rigorous evaluation [42]. Given the global concern over antibiotic resistance, probiotics, as a potential alternative to antibiotics, should not harbor transferable antibiotic resistance genes [43]. In this study, *L. acidophilus* L1 was sensitive to penicillin, ampicillin, chloramphenicol, cefazolin and erythromycin, while resistant to vancomycin, norfloxacin, ciprofloxacin and gentamicin. Importantly, resistance per se is not a safety concern; the risk arises only when resistance genes are transferable between microorganisms [44]. Hemolytic activity is another important indicator for performing in vitro safety assessment of strains. Barzegar et al. [45] reported that *Lactobacillus acidophilus* had no hemolytic activity. Similarly, in the present study, strain L1 showed no hemolysis, suggesting that it either lacks hemolysis-related virulence factors or that such factors are not expressed phenotypically. In addition to safety, the ability to adhere to intestinal epithelial cells is essential for colonization and function of probiotics within the host. Caco-2 cells, as a well-established model of intestinal epithelial cells, are commonly used to evaluate the adhesion ability of potential probiotic strains, enabling the in vitro screening of candidate probiotics [46]. Strain L1 exhibited an adhesion rate exceeding 38% in Caco-2 cell, supporting its strong adhesive capacity and potential as a host-derived probiotic.

Furthermore, this study also evaluated the in vivo safety of strain L1. During the testing period, no diarrhea, illness, or mortality was observed in mice, indicating its safety in an animal model. Mice receiving L1 exhibited dose-dependent reductions in body weight relative to controls, yet these differences lacked statistical significance across the three dosage groups. Numerous *Lactobacillus acidophilus* strains demonstrate a notable anti-obesity effect. Previous studies have demonstrated anti-obesity effects for *L. acidophilus* strains YL01 and *L. acidophilus* NS1 [47,48]. This preliminary finding suggests that L1 may also have potential for weight management, warranting further investigation. The detection of serum BUN, CRE, AST, ALT and TG levels in mice revealed no significant differences in liver and kidney function between those fed *L. acidophilus* L1 and the control group. This finding aligns with reports by Bernardeau et al. [49], which also demonstrated that supplementation with *Lactobacillus acidophilus* had no adverse effects on the internal organs of mice. Regarding antioxidant capacity, oral administration of L1 significantly elevated serum SOD and CAT levels, two key enzymes that play a crucial role in scavenging reactive oxygen species (ROS) and mitigating inflammation [50]. Concurrently, the levels of MDA were significantly lower in the high-dose group fed with *L. acidophilus* L1 compared to the control group. MDA, a byproduct of lipid peroxidation, serves as a reliable biomarker for assessing oxidative stress in organisms [51]. These results indicate that feeding *L. acidophilus* L1 can enhance the antioxidant capacity of mice, which is consistent with the strain’s in vitro free radical scavenging ability. Overall, oral administration of L1 was safe and conferred antioxidant benefits in vivo. Canine-derived probiotics demonstrate beneficial effects on the gastrointestinal microbiome and immune system of various species [52]. For example, supplementation with canine-derived LAB increased the abundance of *Lactobacillus* [53], while Zhao et al. [31] demonstrated that *Pediococcus acidilactici* GLP06 supplementation restructured the canine gut microbiota and improved intestinal health in Beagle models. A future research focus will be the influence of probiotic strains on the gut microbial ecosystem of puppies.

The whole genome of *L. acidophilus* L1 was sequenced to elucidate its potential biological functions. The genome size of *L. acidophilus* L1 was 2,106,895 bp, with an average GC content of 34.99%, had a similar genomic GC content and genome size compared with other reported strains [54]. Based on COG, GO, and KEGG functional annotations, we identified key genes associated with global metabolic pathways, including carbohydrate metabolism, membrane transport, translation, and nucleotide metabolism. In the COG database, the most abundant gene category was Translation, ribosomal structure, and biogenesis (J;190 genes), followed by Carbohydrate transport and metabolism (G;181 genes), consistent with the genomic features of *L. acidophilus* GLA09 [55]. Genetic analysis revealed that 1389 protein-coding genes were functionally annotated in the GO database, predominantly classified under Molecular Function. Within this category, the genes were primarily enriched in ATP binding and DNA binding. Through efficient energy metabolism, it sustains stress tolerance and rapidly adapts to environmental changes via precise gene regulation. This feature provides a molecular basis for its colonization advantage and beneficial function [56,57]. KEGG analysis showed significant enrichment in the Global and Overview Maps and Carbohydrate Metabolism pathways, further highlighting the L1′s core metabolic characteristics and environmental adaptation strategies [58]. The gene analysis revealed many common carbohydrate metabolism-related genes in *L. acidophilus* strains. Carbohydrate-active enzyme (CAZyme) analysis identified 90 genes, including 44 glycoside hydrolases (GHs) and 22 glycosyltransferases (GTs), which are critical for carbohydrate metabolism and survival in the gut. This property lays a molecular foundation for the development of targeted probiotic agents, such as those with antidiarrheal and immunomodulatory effects [59]. The L1 genome harbors two antibiotic resistance genes, *lnuC* and *tetW,* both located on the chromosome rather than plasmids, thereby reducing the risk of horizontal gene transfer. In addition, VFDB annotation identified 181 virulence factor-associated genes, of which 7 exhibited >60% sequence similarity (Appendix A). Notably, many virulence-associated genes are not intrinsically pathogenic but encode functions essential for probiotic fitness; these may be regarded as health-promoting determinants [60]. Nevertheless, comprehensive safety profiling remains essential before clinical or industrial application, as virulence and resistance genes are central to probiotic safety evaluation [22]. Additionally, the genome of L1 contains two CRISPR sequences, which can limit the spread of antimicrobial resistance genes and provide the potential for defense against incoming extrachromosomal DNA molecules [61]. Choloylglycine hydrolase (CBH), hydrolyzing glycine-bile acid amide bonds, and inorganic pyrophosphatase, modulating bile acid coenzyme A activity [48], were identified in the L1 genome. This suggests a potential mechanism underlying the strain’s bile salt tolerance. Moreover, L1 also harbors genes related to bacteriocin production and adhesion, correlating with the in vitro test results.

## 5. Conclusions

In this study, a strain of *L. acidophilus* L1 was successfully isolated from the gastrointestinal tract of a 2-Month-Old Shiba Inu Puppy.

In vitro experiments revealed that L1 was tolerant of acid and bile salts. It exhibited antibacterial activity against five pathogens, displayed free-radical scavenging capacity in vitro, and demonstrated the ability to adhere to intestinal epithelial cell models. In vivo studies showed that L1 enhances antioxidant capacity without toxicity. Genome analysis confirmed its safety and probiotic potential. These findings position *L. acidophilus* L1 as a novel host-derived probiotic candidate for functional companion animal foods.

## Figures and Tables

**Figure 1 microorganisms-13-02095-f001:**
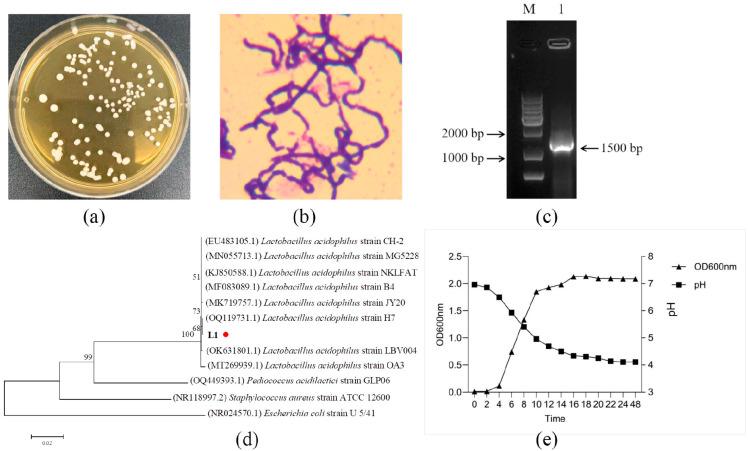
Morphology, staining, electrophoresis, phylogenetic analysis, and growth characteristics of strain L1. (**a**) Colony morphology of strain L1 on MRS agar. (**b**) Gram staining results of strain L1 (×1000). (**c**) The electrophoretic detection of PCR. Line M, 1 Kb DNA ladder I (Novoprotein, DM007), Line 1, 16S rRNA band of strain L1. (**d**) Neighbor-joining phylogenetic tree of strain L1 based on 16S rRNA gene sequences. The red dot is used to highlight the strain L1. (**e**) Growth curve and pH changes in strain L1 in MRS broth over a 48 h incubation period.

**Figure 2 microorganisms-13-02095-f002:**
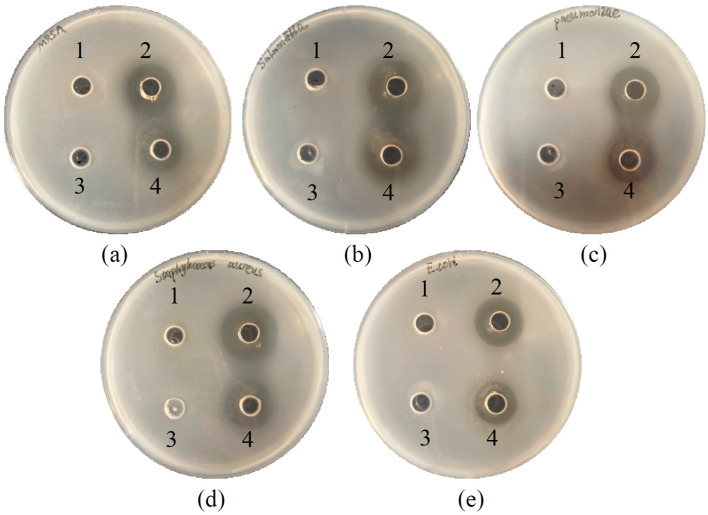
Inhibitory activity of Strain L1 against pathogenic indicator bacteria. The tested pathogens included: (**a**) MRSA. (**b**) *S. typhimurium*. (**c**) *K. pneumoniae*. (**d**) *S. aureus* and (**e**) *E. coli*. For each pathogen, four treatments were applied to LB agar plates: (1) was added to the MRS broth. (2) was added to the cell-free supernatant. (3) was added to the bacterial pellet. (4) was added to the bacterial suspension.

**Figure 3 microorganisms-13-02095-f003:**
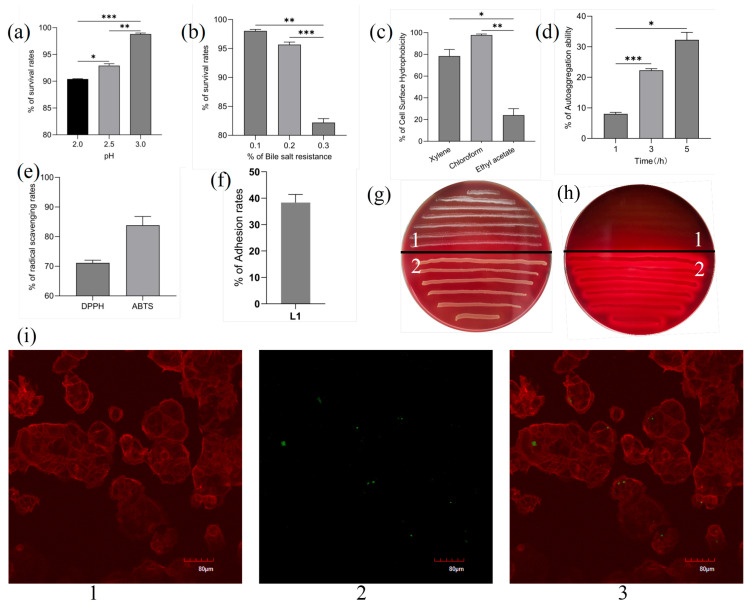
In vitro assessment of *L. acidophilus* L1 probiotic properties. (**a**) Acid tolerance. (**b**) Bile salt resistance. (**c**) Cell surface hydrophobicity. (**d**) Auto-aggregation capacity. (**e**) Radical scavenging activity. (**f**) Adhesion to Caco-2 monolayers. (**g**) Hemolysis assay (blood agar front): 1. *L. acidophilus* L1; 2. *S. aureus* (β-hemolytic control). (**h**) Hemolysis assay (blood agar reverse): 1. *L. acidophilus* L1; 2. *S. aureus*. (**i**) Confocal microscopy of L1 adhesion: (**1**) Actin cytoskeleton (Tracker Red-594); (**2**) FITC-labeled L1; (**3**) Merged image. Statistical significance: * *p* < 0.05, ** *p* < 0.01, *** *p* < 0.001.

**Figure 4 microorganisms-13-02095-f004:**
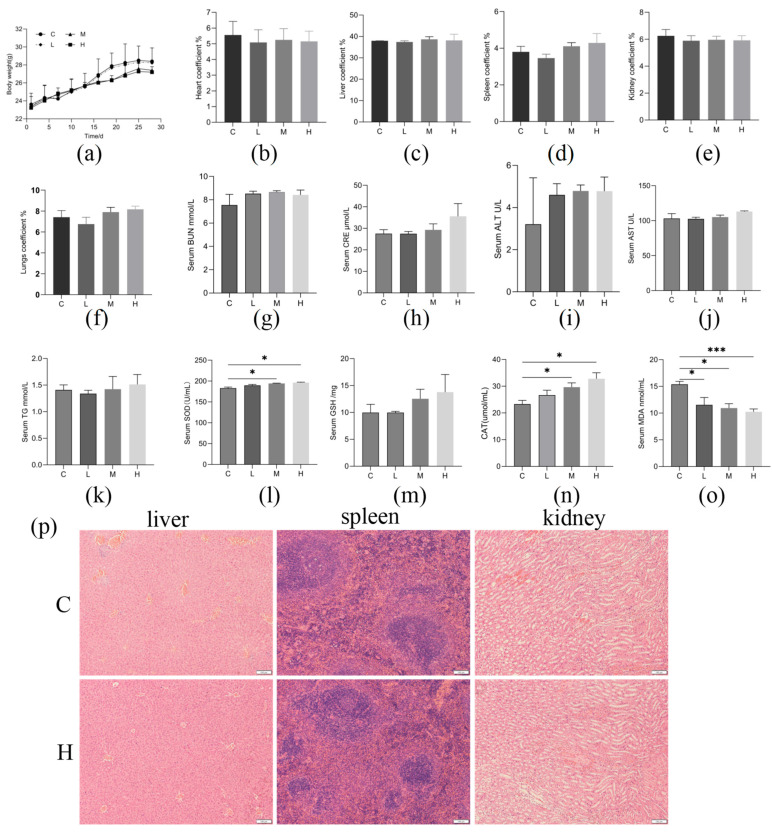
*L. acidophilus* L1 safe evaluation in vivo. (**a**) Body weight. (**b**) heart coefficient. (**c**) liver coefficient. (**d**) spleen coefficient. (**e**) kidney coefficient. (**f**) lungs coefficient. (**g**) blood urea nitrogen. (**h**) Creatinine. (**i**) Alanine aminotransferase. (**j**) Aspartate aminotransferase. (**k**) Triglyceride. (**l**) superoxide dismutase. (**m**) glutathione peroxidase. (**n**) Catalasein. (**o**) Malondialdehyde in mice. (**p**) Representative hematoxylin and eosin-stained pictures of the liver, spleen and kidney (scale bars, 100 μm). Statistical significance: * *p* < 0.05, *** *p* < 0.001.

**Figure 5 microorganisms-13-02095-f005:**
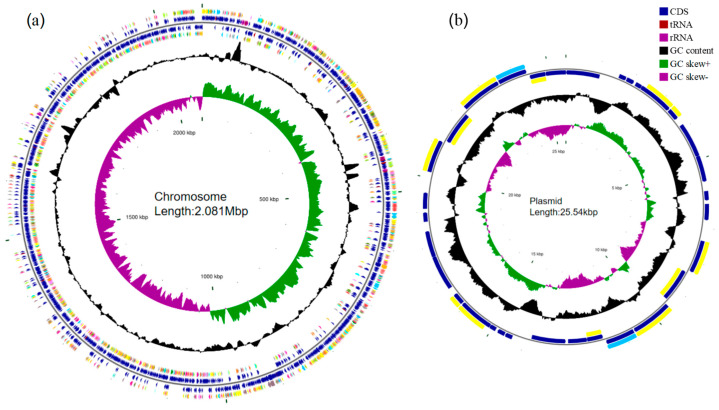
Complete genome map of strain L1. (**a**) The complete circular genome map of strain L1. (**b**) The complete circular plasmid map of strain L1. Circles 2 and 3 (blue) indicate forward and reverse strands, which represent genes for CDS, tRNA, and rRNA. Circle 5 (black) indicates the GC percentage of the genome. Circle 6 (purple and green) represents GC skew.

**Figure 6 microorganisms-13-02095-f006:**
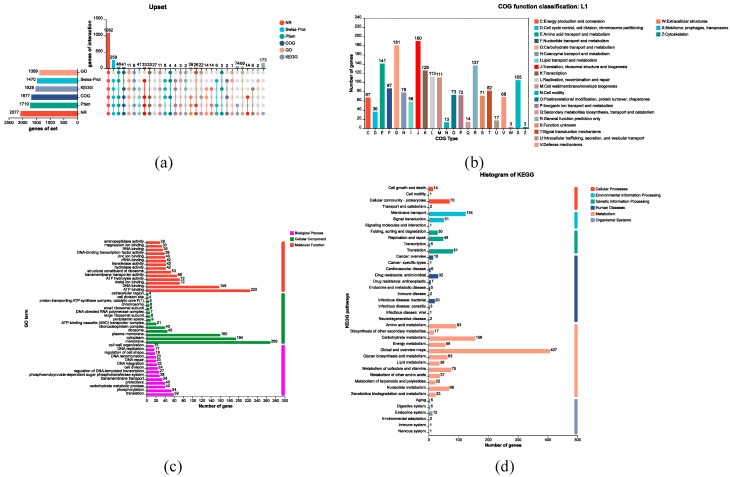
Genomic annotation profiles of *L. acidophilus* L1. (**a**) Annotation statistics across databases. In the central matrix, a single dot denotes a gene uniquely annotated in its corresponding database (listed in the left column). Lines that connect multiple dots signify genes that are annotated in multiple databases. (**b**) COG functional classification of encoded proteins. (**c**) Gene Ontology (GO) term distribution. (**d**) KEGG pathway annotation.

**Figure 7 microorganisms-13-02095-f007:**
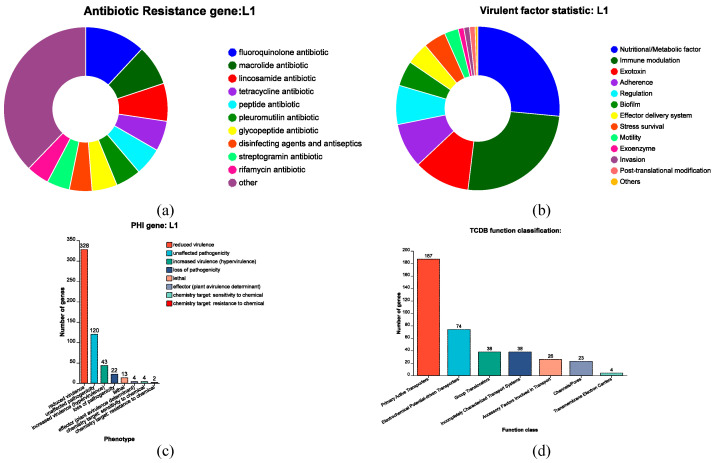
Proprietary database annotations of *L. acidophilus* L1. (**a**) antibiotic resistance. (**b**) virulent factor. (**c**) Pathogen host interactions annotations. (**d**) TCDB transporter protein.

**Table 1 microorganisms-13-02095-t001:** Antibiotic resistance in strain L1.

Antimicrobial Classes	Antimicrobial Agents	Disk Dose (μg)	Inhibition Zone Diameters/mm (IZD) *		
			≤15 mm (R)	16–20 mm (I)	≥21 mm (S)
β-lactams antibiotics	Penicillin	10			28.65 ± 1.06 ^S^
	Ampicillin	10			23.10 ± 0.85 ^S^
Glycopeptides	Vancomycin	30	X ^R^		
Broad-spectrum antibiotics	Chloramphenicol	30			26.95 ± 0.28 ^S^
Quinolone antibiotics	Norfloxacin	10	X ^R^		
	Ciprofloxacin	5	X ^R^		
Aminoglycosides antibiotics	Gentamicin	10	X ^R^		
Cephalosporin antibiotics	Cefazolin	30			35.05 ± 1.49 ^S^
Macrolides	Erythromycin	15			23.50 ± 0.82 ^S^

* R, Resistant; I, Intermediate; S, Sensitive; X, No inhibition zone observed. Values are mean with SDs of three replications.

**Table 2 microorganisms-13-02095-t002:** Genomic features of the *L. acidophilus* L1.

Indicator L1	Number or Content
Chromosome (bp)	2,081,354
Plasmid (bp)	25,541
G + C content of chromosome (%)	34.97
G + C content of plasmid (%)	36.10
Genome (bp)	2,106,895
Gene number	2178
Gene total length (bp)	1,846,086
Gene average length (bp)	879.93
Gene/Genome (%)	87.62
Number of coding sequences	2098
t RNA	65
r RNA	15
Pseudogene number	260
Gene islands	10
Prophages	2

**Table 3 microorganisms-13-02095-t003:** CAZymes-encoding genes of *L. acidophilus* L1.

CAZymes Class Definition	Gene Counts
Auxiliary Activities	7
Carbohydrate-Binding modules	2
Carbohydrate Esterases	15
Glycoside Hydrolases	44
Glycosyl Transferases	22

**Table 4 microorganisms-13-02095-t004:** The key functional genes associated with probiotic characteristics in *L. acidophilus* L1.

Gene ID	Gene Name	Gene Description	Predicted Function
gene0138	*nicD*	alpha/beta hydrolase	Bile salt hydrolase
gene0045	*-*	SGNH/GDSL hydrolase family protein	
gene0151	*-*	alpha/beta hydrolase	
gene1323	*cbh*	choloyglycine hydrolase	
gene1844	*tesA*	SGNH/GDSL hydrolase family protein	
gene0092	*-*	helveticin	Bacteriocin
gene0091	*-*	helveticin	
gene0417	*-*	helveticin J family class III bacteriocin	
gene0177	*-*	helveticin	
gene0565	*-*	Class III bacteriocin	
gene0532		Bacteriocin immunity protein	
gene1668	*-*	helveticin J family class III bacteriocin	
gene1626	*-*	bacteriocin immunity protein	
gene1992	*-*	bacteriocin immunity protein	
gene1993	*-*	bacteriocin immunity protein	
gene0181	*acm*	surface layer protein	Adhesion
gene0185	*acm*	SLAP domain-containing protein	
gene0229	*-*	SLAP domain-containing protein	
gene0230	*-*	SLAP domain-containing protein	
gene0094	*-*	SLAP domain-containing protein	
gene0531	*-*	Putative mucus binding protein	
gene0664	*fnbA*	Mucus-binding protein	
gene0729	*bapA*	LPXTG cell wall anchor domain-containing	
gene0730	*bapA*	Hypothetical protein	
gene0528	*bapA*	Hypothetical protein	
gene0728	*bapA*	Mucin binding domain	
gene0731	*bapA*	Muc B2-like domain	
gene1051	*acm*	SLAP domain	
gene1148	*-*	SLAP domain	
gene1182	*-*	SLAP domain	
gene1184	*bapA*	MucBP domain	
gene1312	*bapA*	mucus-binding protein	
gene1822	*fnbA*	mucus-binding protein	
gene1823	*fnbA*	hypothetical protein	
gene1826	*clfB*	cell surface protein	
gene1827	*clfB*	cell surface protein/YSIRK-type signal peptide-containing protein	
gene1940	*bapA*	BspA family leucine-rich repeat surface protein	
gene1938	*-*	BspA family leucine-rich repeat surface protein	

## Data Availability

The 16S rRNA sequence generated in this study has been deposited in GenBank under accession No. PV335979. The raw sequence data have been deposited in the NCBI database under accession number PRJNA1262664.

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
