# Peer review of "A Novel Lactobacillus acidophilus Strain Isolated from a 2-Month-Old Shiba Inu: In Vitro Probiotic Evaluation Safety Assessment in Mice and Whole-Genome Sequencing Analysis"

_microorganisms, 2025, doi:10.3390/microorganisms13092095_

Round 1
Reviewer 1 Report
Comments and Suggestions for Authors
REVIEW
Dear authors,
The work shows evidence of the characterization of a microorganism with probiotic potential isolated from a Shiba Inu puppy, specifically Lactobacillus acidophilus L1, which meets the basic criteria of molecular identification, in vitro evaluation such as high percentages of survival and adherence to intestinal epithelial cells, as well as evaluation in an in vivo model in mice where it demonstrates its safety under the established conditions of the treatment carried out (bacterial concentration and treatment time). The results found propose that L. acidophilus L1 be considered as a potential probiotic for use in the health of pets (dogs) since the microorganism is species-specific.
Please consider the following comments to improve the content of your manuscript before publication.
- The initial weight of the mice is not indicated in section 7 Safety evaluation of the L1 strain in mice.
- What was the reason for using ICR mice?
- How to establish the amount of inocula administered in the treatment (28 days)?
- In section 3 Antimicrobial ability they only place Figure 2, however the measurements of the diameters of the halos are not indicated, I consider that they should place a table or graph the values of the diameters so that the result is quantitative.
- Adjust the text on lines 328 to 347.
- To complement the evaluation of the % coefficient of the heart, liver, spleen and kidneys, it is necessary to perform histopathology (HE staining) from these organs; although the percentages do not show damage, evidence of absence at the tissue level must be demonstrated.
- They did not take into account the main organ in the animal model, the intestine, although the survival percentage in the in vitro model is greater than 90% when it is transferred to the in vivo model there are multiple factors that affect the survival of microorganisms with probiotic potential, I consider that they should complement the work carrying out survival in mice through kinetics by taking the points each week.
- Although the metabolic parameters turned out to be favorable, the intestinal microbiota plays a very important role, which does not mean that L1 is directly responsible for the regulation. It has been shown that some probiotic strains can stimulate the growth or production of metabolites in native bacterial genera with probiotic potential, so it is necessary to perform a scan of the intestinal microbiota or at least the intentional isolation of the Lactobacillus genus by traditional culture techniques.
- Lines 119 and 129: write the unit correctly “mL”.
- Line 219: write the term in italics “ad libitum”.
- Lines 489 and 496: write the term in italics “in vitro”.
Please amend the requested comments and submit the revision file.

Author Response
comment 1. [The initial weight of the mice is not indicated in section 7 Safety evaluation of the L1 strain in mice.]
Response: [Thank you very much for the suggestion. This study employed SPF-grade ICR mice that were 4 weeks old and weighed 20–23 g; these details have been incorporated into the revised manuscript,as seen in section 2.7.]
comment 2. [What was the reason for using ICR mice?]
Response: [Thank you very much for the comments.We selected ICR mice for the probiotic safety assessment based on the following rationale:
(1) Their docile temperament, moderate size, and ease of handling facilitate experimental procedures.
(2) Their commercial availability and cost-effectiveness ensure logistical feasibility.
(3) Critically, a substantial body of literature supports their established use in probiotic functionality research.]
comment 3. [How to establish the amount of inocula administered in the treatment (28 days)?]
Response: [Thank you very much for the comments. As reported in Frontiers in Microbiology (2024;15:1369402), the canine study employed oral doses of 2 × 10⁸–2 × 10¹⁰ CFU/day/dog—that aligns exactly with the 10⁷–10⁹ CFU/day/mouse range used in ICR-mouse safety protocols. Treatments were administered for four weeks following a one-week adaptation period.]
comment 4. [In section 3 Antimicrobial ability they only place Figure 2, however the measurements of the diameters of the halos are not indicated, I consider that they should place a table or graph the values of the diameters so that the result is quantitative.]
Response: [Thank you very much for the suggestion. The precise antibacterial zone diameters have now been tabulated and provided in the supplementary materials, as seen in Table S1.]
comment 5. [Adjust the text on lines 328 to 347.]
Response: [Thank you very much for the suggestion. The revised manuscript has been thoroughly revised and refined for clarity and accuracy.]
comment 6. [To complement the evaluation of the % coefficient of the heart, liver, spleen and kidneys, it is necessary to perform histopathology (HE staining) from these organs; although the percentages do not show damage, evidence of absence at the tissue level must be demonstrated.]
Response: [Thank you very much for the suggestion. The updated manuscript now incorporates the corresponding HE staining results, as seen in Figure 4p.]
comment 7. [They did not take into account the main organ in the animal model, the intestine, although the survival percentage in the in vitro model is greater than 90% when it is transferred to the in vivo model there are multiple factors that affect the survival of microorganisms with probiotic potential, I consider that they should complement the work carrying out survival in mice through kinetics by taking the points each week.]
Response: [We sincerely appreciate your thorough evaluation of our work and the valuable suggestion concerning the implementation of survival studies in mice. We strongly agree that this experiment is essential for further validating strain functionality. However, all experimental animals have been humanely euthanized in this study phase, and biological samples are no longer available. Consequently, we are unable to incorporate this specific analysis in the current revision. In subsequent research, we will utilize canine models to systematically evaluate the strain's in vivo survival kinetics and colonization capacity through integrated qPCR quantification and selective culture methodologies. The results of these experiments will be reported in a forthcoming publication and will allow us to directly correlate the in vitro survival rate demonstrated here with in vivo persistence.]
comment 8. [Although the metabolic parameters turned out to be favorable, the intestinal microbiota plays a very important role, which does not mean that L1 is directly responsible for the regulation. It has been shown that some probiotic strains can stimulate the growth or production of metabolites in native bacterial genera with probiotic potential, so it is necessary to perform a scan of the intestinal microbiota or at least the intentional isolation of the Lactobacillus genus by traditional culture techniques.]
Response: [Thank you very much for the suggestion. We fully agree that perform a scan of the intestinal microbiota would provide additional insight. However, owing to current the tight deadline of the revision, we were unable to perform the requested additional experiments in this revision. We have integrated the reviewers' suggested experiments into our next research phase, utilizing companion animal models (dogs) to systematically evaluate probiotic effects on juvenile intestinal microbiota. This expanded approach will validate the broader applicability of our findings while generating translational evidence for probiotic use in veterinary practice. We sincerely appreciate your thorough assessment and constructive feedback on our work.]
comment 9. [Lines 119 and 129: write the unit correctly “mL”.]
Response: [We are sorry for our careless mistakes.The requested revisions have been incorporated into the revised manuscript.]
comment 10. [Line 219: write the term in italics “ad libitum”.]
Response: [We are sorry for our careless mistakes. The requested revisions have been incorporated into the revised manuscript.]
comment 11. [Lines 489 and 496: write the term in italics “in vitro”.]
Response: [We are sorry for our careless mistakes. The requested revisions have been incorporated into the revised manuscript.]
Reviewer 2 Report
Comments and Suggestions for Authors
Manuscript ID: microorganisms-3776320
Title: A Novel Lactobacillus acidophilus Strain Isolated from a 2 - Month - Old Shiba Inu: In Vitro Probiotic Evaluation, Safety Assessment in Mice and Whole - Genome Sequencing Analysis
In this study, the authors isolated a strain of Lactobacillus acidophilus (designated L1) from the gastrointestinal tract of a 2-Month-Old Shiba Inu Puppy. The probiotic potential and safety were analyzed in vivo and in vitro. It’s a very interesting study. Only few questions should be addressed to make it optimized.
- Figure 1: The DNA Marker used in agarose gel electrophoresis should clearly specify the fragment sizes and product information (e.g., manufacturer and catalog number).
- How was the dose for the in vivo study determined? The body weights of mice in the medium- and high-dose groups were 1-2g lower than those in the control and low-dose groups, with the low-dose group also slightly lower than the control (Figure 4a). Would this indicate an adverse effect on mouse growth? Or the dose used in this study be suboptimal? Moreover, how is the dose-dependent increase in spleen and lung organ indices and serum biomarkers (urea nitrogen/BUN, ALT, TG) to be interpreted?
Author Response
comment 1. [Figure 1: The DNA Marker used in agarose gel electrophoresis should clearly specify the fragment sizes and product information (e.g., manufacturer and catalog number).]
Response: [Thank you very much for the suggestion. The manufacturer and catalog number of the DNA marker have been added to the revised manuscript, as seen in Figure 1c.]
comment 2. [How was the dose for the in vivo study determined? The body weights of mice in the medium- and high-dose groups were 1-2g lower than those in the control and low-dose groups, with the low-dose group also slightly lower than the control (Figure 4a). Would this indicate an adverse effect on mouse growth? Or the dose used in this study be suboptimal? Moreover, how is the dose-dependent increase in spleen and lung organ indices and serum biomarkers (urea nitrogen/BUN, ALT, TG) to be interpreted?]
Response: [Thank you very much for the suggestion. Although the final mean body weight of the probiotic group was 1–2 g lower than that of the control group, this difference was not statistically significant(p>0.05). Importantly, the baseline body weight of the probiotic group was already ~1 g lower. Taken together, these data indicate that oral administration of the probiotic had no appreciable effect on body weight reduction. There were no statistically significant differences in spleen and lung organ indices or serum biomarkers (urea nitrogen/BUN, ALT, TG) across different doses, and therefore, a dose-dependent effect could not be directly established. To determine the dose-response effects of probiotics, we will continue examining inflammatory factors associated with hepatic and renal function in mice in future.]
Reviewer 3 Report
Comments and Suggestions for Authors
This manuscript reports the isolation, probiotic characterization, safety assessment, and genomic analysis of a novel Lactobacillus acidophilus strain (L1) from a puppy. It presents a comprehensive study with both in vitro and in vivo evaluations and whole-genome sequencing. The work is timely and relevant, especially given the increasing interest in host-derived probiotics for companion animals.
While the study presents a thorough evaluation of Lactobacillus acidophilus L1 as a potential host-derived probiotic, several scientific limitations should be addressed before publication.
- The absence of a benchmark or positive control strain limits the ability to contextualize the probiotic potential of L1 relative to established strains.
- Although antibiotic susceptibility was assessed phenotypically, resistance to vancomycin, ciprofloxacin, norfloxacin, and gentamicin raises concerns regarding the presence of transmissible resistance genes. Genomic screening for mobile genetic elements and resistance determinants (e.g., via ResFinder or CARD databases) is necessary to assess biosafety more comprehensively.
- The adhesion assay relies on Caco-2 cells—a human intestinal model—which may not fully reflect the canine gastrointestinal environment. This species mismatch should be acknowledged, and future studies should consider using canine-derived epithelial models.
- The in vivo assessment focuses narrowly on antioxidant activity without verifying gut colonization or evaluating broader probiotic effects on microbiota composition or host physiology. Fecal recovery of the strain and microbiome profiling would provide essential mechanistic insight.
- While whole-genome sequencing is included, key functional genes associated with probiotic traits (e.g., bile salt hydrolases, bacteriocins, adhesion proteins) are not annotated or discussed. These omissions, while not fatal, should be addressed or transparently acknowledged to enhance the scientific rigor and translational relevance of the study.
Author Response
comment 1. [The absence of a benchmark or positive control strain limits the ability to contextualize the probiotic potential of L1 relative to established strains.]
Response: [Thank you very much for the suggestion. In the Discussion, we therefore benchmarked our findings against the documented functions of established probiotic strains, and we will continue to track emerging data on puppy-derived probiotics.]
comment 2. [Although antibiotic susceptibility was assessed phenotypically, resistance to vancomycin, ciprofloxacin, norfloxacin, and gentamicin raises concerns regarding the presence of transmissible resistance genes. Genomic screening for mobile genetic elements and resistance determinants (e.g., via ResFinder or CARD databases) is necessary to assess biosafety more comprehensively.]
Response: [Thank you very much for the suggestion. In this revision, we have incorporated additional data analyses to further substantiate our findings,as seen in Figure7a and Table S2.]
comment 3. [The adhesion assay relies on Caco-2 cells—a human intestinal model—which may not fully reflect the canine gastrointestinal environment. This species mismatch should be acknowledged, and future studies should consider using canine-derived epithelial models.]
Response: [We sincerely appreciate the reviewers’ insightful comments. The reviewers’ recommendations will guide our immediate follow-up investigations.]
comment 4. [The in vivo assessment focuses narrowly on antioxidant activity without verifying gut colonization or evaluating broader probiotic effects on microbiota composition or host physiology. Fecal recovery of the strain and microbiome profiling would provide essential mechanistic insight.]
Response: [Thank you very much for the suggestion. Owing to current experimental time limitations, we were unable to perform the requested additional experiments in this revision. We have incorporated pertinent literature in Discussion to strengthen our conclusions and have explicitly acknowledged the limitations of the present study. We have integrated the reviewers' suggested experiments into our next research phase, utilizing companion animal models (dogs) to systematically evaluate probiotic effects on juvenile intestinal microbiota.]
comment 5. [While whole-genome sequencing is included, key functional genes associated with probiotic traits (e.g., bile salt hydrolases, bacteriocins, adhesion proteins) are not annotated or discussed. These omissions, while not fatal, should be addressed or transparently acknowledged to enhance the scientific rigor and translational relevance of the study.]
Response: [Thank you very much for the suggestion. In this revision, we have incorporated additional data analyses to further substantiate our findings, as presented in Table 4.]
Round 2
Reviewer 2 Report
Comments and Suggestions for Authors
The author addressed some of my questions, but I am not entirely convinced by their explanation. Additionally, I did not find the mention of " the final mean body weight of the probiotic group was 1–2 g lower than that of the control group," from response in figure 4a. And the author did not address whether the dosage was appropriate. Based on the results, the low dose may not be the optimal choice for application, and the optimal dose might be even lower than the low dose used. Therefore, I suggest that the author carefully verify this and provide a proper explanation.
Author Response
Comment1:[The author addressed some of my questions, but I am not entirely convinced by their explanation. Additionally, I did not find the mention of " the final mean body weight of the probiotic group was 1–2 g lower than that of the control group," from response in figure 4a. And the author did not address whether the dosage was appropriate. Based on the results, the low dose may not be the optimal choice for application, and the optimal dose might be even lower than the low dose used. Therefore, I suggest that the author carefully verify this and provide a proper explanation.]
Response: [Thank you very much for the comments. The probiotic gavage dosage was determined in accordance with established protocols from previously published literature, which has been cited in line 230 of the 2.7 section(Front. Microbiol. 2024;15:1369402). We are therefore confident that the administered dosage is appropriate. Concerning the observed reduction in body weight, we have further addressed this point in Line 348-350 of the section 3.6 “The body weights of mice treated with L1 across all three groups exhibited dose-dependent reductions compared to the control group, with greater weight loss observed at higher doses(Figure 4a).” and Line 518-524 of the Discussion “Mice receiving L1 exhibited dose-dependent reductions in body weight relative to controls, yet these differences lacked statistical significance across the three dosage groups. Numerous Lactobacillus acidophilus strains demonstrate a notable anti-obesity effect. Previous studies have demonstrated anti-obesity effects for L. acidophilus strains YL01 and L. acidophilus NS1[47,48]. This preliminary finding suggests that L1 may also have potential for weight management, warranting further investigation.” , supported by citations from References 47 (Int. J. Biol. Macromol. 2025;300:140287.)and 48(J. Endocrinol. 2018;237(2):87-100.). We hope that the reviewers find our explanations satisfactory and accept our conclusions.]